# Root Exudates in Soilless Culture Conditions

**DOI:** 10.3390/plants14030479

**Published:** 2025-02-06

**Authors:** Brechtje de Haas, Emmy Dhooghe, Danny Geelen

**Affiliations:** Department of Plants and Crops, Faculty of Bioscience Engineering, Ghent University, Coupure Links 653, 9000 Gent, Belgium; brechtje.dehaas@ugent.be (B.d.H.); emmy.dhooghe@ugent.be (E.D.)

**Keywords:** soilless cultivation, hydroponics, horticulture, PGPB

## Abstract

Root metabolite secretion plays a critical role in increasing nutrient acquisition, allelopathy, and shaping the root-associated microbiome. While much research has explored the ecological functions of root exudates, their relevance to horticultural practices, particularly soilless cultivation, remains underexplored. Steering root exudation could help growers enhance the effectiveness of plant growth-promoting bacteria. This review summarizes current knowledge on root exudation in soilless systems, examining its process and discussing environmental influences in the context of soilless cultivation. Plants in soilless systems exhibit higher total carbon exudation rates compared to those in natural soils, with exudation profiles varying across systems and species. Root exudation decreases with plant age, with most environmental adaptations occurring during early growth stages. Several environmental factors unique to soilless systems affect root exudation. For instance, nutrient availability has a major impact on root exudation. Light intensity reduces exudation rates, and light quality influences exudation profiles in a species- and environment-dependent manner. Elevated CO_2_ and temperature increase exudation. Factors related to the hydroponic nutrient solution and growing media composition remain insufficiently understood, necessitating further research.

## 1. Introduction

Root exudation refers to metabolite secretion from plant roots. It is defined as “plant-derived primary and secondary metabolites of both low molecular weight (<1000 Da; e.g., sugars, organic acids, phenolics, vitamins) and high molecular weight (>1000 Da; e.g., enzymes, mucilage)” [1]. This process is part of the broader category of rhizodeposition, which concerns the release of all organic C from roots and includes the loss of border cells (or border-like cells).

Since root exudates are underground, can be used by microbes, and may be absorbed by the soil, different methods have been developed to measure them [1]. These methods differ by several factors: (1) whether root exudates are collected in soil, hydroponics, or a hybrid system, (2) whether exudates are collected destructively or non-destructively (leachate collection), (3) whether whole plants or (a part of) the root system is used, (4) whether the environment is sterile or non-sterile, (5) whether the sampling solution is based on nutrient solution, CaCl_2_, or water, (6) the purity of the water, (7) whether specialized microcups are used, (8) the duration of root sampling, (9) and whether the roots are washed, and if so, (10) whether a recovery period is included. Depending on the method used, different exudate compositions have been identified [2], opening the discussion which of these methods most closely reflect the biological reality.

The biological reality will depend on how plants are grown and requires the appropriate method of analysis. In soilless horticulture, plants are not grown on natural soils, but in substrate-based systems, including the nutrient film technique (NFT), the deep flow technique (DFT), and drip irrigation systems. In all these systems, at least part of the roots is supported by a substrate block. These substrates can be as diverse as from rockwool to organic peat. In a soilless system, the nutrient solution and incubation conditions are highly controlled, which allows the evaluation of the impact of environmental factors on root exudation.

Natural soil has a major impact on root growth and the root microbiome, and it contributes to shaping the quality and quantity of root exudate [3,4]. Root exudation is linked to root morphology [5,6,7]. In soilless cultivation, root morphology can vary substantially due to the diverse artificial environments. For example, plants may delay suberization in well-aerated DFT systems [8] and change root morphology in different growing systems [9]. Therefore, the root exudation rate and composition are likely to differ in soilless cultivation compared to those in natural soils.

Root exudation supports various physiological and ecological functions essential for plant survival and growth. For example, it facilitates P acquisition by the solubilization of phosphate [10]. Root exudates also have allelopathic properties that influence the growth and development of neighboring plants [11]. Exudate interacts dynamically with the rhizosphere microbiota, serving both as a nutrient source and as a repellent to certain microorganisms [12,13]. Furthermore, root exudates contribute to plant stability by promoting the formation of soil aggregates, thus reinforcing root anchorage [14]. In soilless horticulture, the nutrient acquisition and allelopathy functions of exudate are obsolete. Growers add nutrients in a plant-available form that provide plants with optimal conditions for growth. However, the selection of (potentially beneficial) microbes is of interest to the grower [10].

In this review, we summarize insights on root exudation in soilless systems and discuss the process of root exudation. Building on this knowledge, we explore the implications of environmental factors for root exudation in soilless horticulture.

## 2. Root Exudate in Soilless Systems

Although the primary objective of root exudate research is to elucidate the mechanisms of root exudation in natural soil environments, most studies have been conducted under controlled conditions [1]. In these experiments, plant roots are immersed in a nutrient solution, frequently without any growing medium. Findings from such studies indicate that most root metabolites also occur in the exudate. Carbohydrates typically constitute a significant proportion of root exudate (>25%), followed closely or occasionally surpassed by organic acids, with amino acids and secondary metabolites forming subsequent fractions [8,9,10,11]. Conversely, lipids, proteins, and enzymes are only sometimes reported as components of root exudates. This may be attributable to the hydrophobic characteristics of lipids, which limit their release into aqueous solutions. Proteins and enzymes, in contrast, may exhibit lower exudation potential and associate more strongly with the root surface [15].

Since soils vary in structure and composition, and various soilless systems are being used, comparative root exudation studies should take into account that the collection of exudate samples is conducted using different methodologies. For example, soil-based systems necessitate a washing step to remove soil, which may influence the exudate profile [16]. Despite these challenges, a few studies have compared root exudates in different systems.

Previous observations suggested that the total carbon (C) exudation rate is higher in hydroponic systems than in natural soils [17]. Furthermore, the exudate profiles differ between systems, with these changes being species-dependent. For example, Lacy phacelia and Egyptian clover exhibited a higher concentration of secondary compounds in their root exudates when grown in natural soils compared to hydroponic systems. White mustard and bristle oat, on the other hand, showed the opposite trend [17]. Similar trends were observed for primary metabolites. Maize, Lacy phacelia, and Egyptian clover released more organic acids and amino acids in soils than in hydroponic systems, whereas bristle oat and white mustard exhibited the reverse response [2,17].

## 3. Process of Root Exudation

At the physical boundary of the root, its function as a barrier for external chemicals and a facilitator of molecule uptake must be seamlessly integrated. While the epidermis and cortex act as a barrier for organic molecules, mineral nutrients can move across the apoplast between the cortex cells and are selectively taken up by the vascular bundle (Figure 1) [18].

The endodermis is the outermost cell layer of the vasculature that is the most crucial layer for selective mineral nutrient uptake [19]. Surrounding the endodermis, a suberin layer prevents the passage of nutrients as it is impermeable to water, allowing access to the endodermis only at specific sites [19]. Another selective barrier is called the Casparian strip, located at the cross-walls between endodermal cells, which is impermeable to charged molecules [20]. Together, these structural features prevent the apoplastic movement of molecules between the root transport vessels and the outer cell layers.

Roots engage in an exchange with the surrounding soil to obtain water and nutrients and to release root exudates; therefore, the barriers in the root structure cannot fully isolate the root from the environment. At the growing root tip, the endodermis is at an immature stage and suberin is gradually laid down beyond the root elongation zone where it becomes a continuous layer. Once the suberin layer is fully formed, some endodermal cells differentiate into passage cells, which lack the suberin layer [19]. The spatiotemporal absence of these barriers is a prerequisite for the exchange of materials between the root and the soil.

To move to and from the vascular bundle, metabolites must traverse the plasma membrane of the endodermis cell. Several transporter families have been implicated in the uptake and release of metabolites (Table 1) [21]. However, many key transporters remain unidentified [22]. The involvement of multiple transmembrane proteins suggests that plants exert a degree of control over the rate and composition of exudation [23].

The final condition for root exudation is the availability of exudates. These metabolites can be synthesized in various plant tissues, and when they accumulate in the roots, they can be secreted. For this to occur, metabolites must be unloaded from the phloem into the pericycle. This unloading is facilitated by specialized funnel plasmodesmata, primarily located at the root tip [24].

Root exudation is mostly a (facilitated) passive process, as there is a large electrochemical gradient between the cytosol and the cell wall space [25]. This gradient is maintained by microorganisms that metabolize the exudate molecules [26].

**Table 1 plants-14-00479-t001:** Transmembrane proteins involved in root exudation.

Transporter Family	(Alleged) Compound Being Transported	Passive or Active Transport	Evidence for Involvement in Exudation	References
UMAMIT	Amino acids	Passive	Strong	[27]
CAT	Amino acids/glutamine	Passive	Present in root tip	[28]
GDU	Glutamine (amino acids?)	Passive	Strong	[29]
SWEET	Hexose and sucrose	Passive	Mostly connected with pathogens/strong microbial interactions	[22,30]
ALMT/malate	Malate/organic acids	Passive	Strong	[31,32]
MATE/citrate	Citrate/organic acids	H^+^-coupled antiport activity	Strong	[32]
ABC	Phenolics	ATP-dependent	Strong	[33]

## 4. Factors That Influence Root Exudation in Soilless Cultivation

Recent advancements have provided insight into the mechanisms underlying root exudation. In building upon this foundational knowledge, a relevant question arises regarding the potential applications of these mechanisms in promoting PGPB in soilless cultivation systems. Specifically, it is important to identify which factors can be influenced by soilless cultivation practices and whether these factors subsequently affect the rates and profiles of root exudation.

### 4.1. Plant Age

Understanding how plant age influences root exudation is essential for determining when root exudation is most adaptive to varying environmental conditions. Root exudation is a dynamic process throughout the plant’s life cycle. Total carbon (C) exudation decreases and the exudate profile changes with plant age [34,35]. In early stages, plants secrete relatively high amounts of sugars, which shift to phenolics and hormones as the plant matures [36,37]. Sugar exudation at an early stage of development may help in establishing a beneficiary root microbiome, while secondary metabolites likely play a role in plant defense mechanisms against pathogens and in allelopathic interactions. The strong exudation potential of young plants accords with a high responsivity to environmental stimuli [35,38]. The reason why young roots have a higher exudation rate is not entirely clear yet. It could be related to the lower suberization level of the roots. The early stages of plant growth are critical for modulating root exudation and optimizing PGPB applications.

### 4.2. Root Physical Environment

The root environment in soilless cultivation can vary widely. Roots may grow in water, as in deep flow technique systems, or in mist, as in aeroponics. They may also be supported by minimal anchoring structures, like small plugs, or be placed in pots. The types of growing media used are diverse, including peat, coco coir, green waste compost, perlite, and rockwool. Even natural soils can induce variations in root exudation profiles [3], suggesting that the unique conditions of soilless systems are likely to impact root exudation as well. These environments differ in both chemical and physical characteristics.

#### 4.2.1. Chemical Factors

Plants are typically watered with a nutrient solution that has a characteristic electrical conductivity (EC), pH, and nutrient composition. EC is a key parameter for growers to monitor the quality of the nutrient solution. EC is frequently used as an indicator for studying salt stress or the accumulation of autotoxins. While plants in greenhouse environments may experience stress due to autotoxins, high levels of sodium (Na+) salts are relatively uncommon. As a result, there is a knowledge gap regarding how variations in EC, driven by the combined presence of nutrients, influence root exudation. This could be tested by applying different EC levels, ranging between 1.0 and 2.5 mS cm^−1^ in a deep flow system, while plants are grown in an inert medium.

In addition to electrical conductivity (EC), pH is another critical parameter determining nutrient availability. By modulating pH, plants can locally enhance the availability of essential nutrients. For instance, organic acids in root exudation lower the rhizosphere pH, releasing nutrients in acid-soluble forms [39].

In the context of soilless cultivation, nutrients and pH are continuously monitored and resupplied when they drop below a threshold. The effect of pH on root exudation is relatively important as some findings show that a low pH reduces malate exudation in soybean [40], while in contrast, a low pH increases the exudation of 3-epi-brachialactone in *Brachiaria humidicola* [41]. More research is needed to establish a general principle regarding the influence of pH on root exudation. For this, several pH levels, ranging between 5.5 and 7, should be tested while plants are grown in an inert medium. Plants with a different pH range might react differently in their exudation strategy to the same pH level, which should be included in experiments.

While the combined effects of nutrients on root exudation have rarely been studied, research on individual nutrients is common. Much of the focus has been on how root exudates enhance nutrient availability in the rhizosphere [10,23]. However, in the context of PGPB applications, the question is reversed: how do nutrient levels influence the quantity and quality of root exudates? Nitrogen addition in different forms typically increases amino acid exudation [42,43] but has a net negative impact on overall root exudation [44]. Phosphorus deficiency induces the exudation of organic acids to solubilize P [43,45]. Potassium deficiency reduces sugar exudation [43], while iron deficiency generally triggers the exudation of organic acids and/or phytosiderophores, depending on the plant’s uptake strategy [45]. However, in soilless cultivation, we expect that the combined effect of nutrients on root exudation via EC and pH is bigger than the impact of each nutrient in isolation.

Growing media vary significantly in sorption characteristics. For instance, clay, with its high sorption capacity, can retain 20–70% of the exudate molecules, while inert materials such as sand and glass beads display a very low sorption capacity. The exudation profile of *Brachypodium distachyon* remains stable under different sorption capacities [9]. The impact of different growing media on root exudation rates remains an area that is poorly studied. A higher sorption capacity results in a larger gradient between the plant cytoplasm and the soil and may therefore increase exudation rates. However, lower sorption may result in the greater leaching of exudates, as they are not adsorbed to the soil, and increase exudation rates.

#### 4.2.2. Physical Factors

Within a single category of growing media, various particle sizes can exist, and the impact of particle size on root growth has been explored [9]. Varying substrate particle size did not affect the root exudation profile of *Brachypodium dystachion*.

Soil porosity, which refers to the space between particles, consists of both air and water porosity. This porosity influences the humidity around the roots, and higher humidity levels are associated with an increase in root hair formation [46]. However, the role of root hairs in exudation remains a topic of debate [47,48,49,50].

Deep water culture, or deep flow technique, is a hydroponic system where plants float in an aerated water basin. Under these conditions, plants may form aerenchyma, a tissue that enables the distribution of oxygen throughout the plant body that is especially important for water-submerged organs [51]. Aerenchyma formation is a response to flooding, where cortical cells undergo programmed cell death to facilitate gas exchange. The impact of aerenchyma formation on root exudation remains unknown. Moreover, in the deep flow technique, root suberization may be either delayed or increased, depending on oxygen levels [8]. Although suberization interacts with the root-associated microbiome [52], its effects on root exudation are still unclear. We hypothesize, though, that a delay or decrease in suberization would increase exudation levels, as it is an important component of the endodermis. This could explain why a higher exudation rate has been recorded in plants grown in hydroponics compared to in a field, as the hydroponic set-up was well aerated and should have resulted in delayed suberization [17].

### 4.3. Aboveground Conditions

The conditions around and near plant roots will likely have a stronger impact on root exudation than aboveground conditions. Nevertheless, aboveground conditions have an effect on root exudation. A part of the assimilated carbon from photosynthesis is allocated to the root and contributes to exudation. Light, temperature, and CO_2_ are critical for photosynthesis capacity and therefore will likely indirectly determine root exudation.

The effect of light intensity on root exudation has been investigated in studies using ambient light and partial shading. For example, clover showed lower amino acid exudation under reduced light intensity, while tomato plants showed less responsiveness [53]. In seagrass species, nitrogen-containing compounds were secreted at lower rates under reduced light [54]. In artificial light conditions, lettuce exhibited a reduced total carbon exudation rate of certain autotoxins at higher light intensities [55]. This suggests that artificial light intensity might have a reverse effect compared to ambient light intensity.

The photoperiod likely influences the rate of root exudation, although only one study has specifically addressed this [56]. Cucumber plants had a lower exudation rate of phytotoxins when the daylength was shortened from 14 to 10 h [56]. The authors did not account for root weight, and the effect was likely due to the reduced root mass in shorter daylengths. More research on the impact of the photoperiod on root exudation is warranted, particularly since light manipulation is commonly used in soilless cultivation to extend daylength. In these studies, the daylength should range up to 18 or 20 h. We hypothesize that a longer artificial photoperiod may have a similar effect to increasing artificial light intensity.

Light quality also plays a significant role in root exudation. For example, blue light promoted more nodulation than red light in soybean. The light signal was transduced through HY5 orthologs and FLOWERING LOCUS T under blue light and full-spectrum light [57]. Both factors are mobile factors moving from the shoot to the root. Interestingly, HY5 has also been implicated in the far-red response in increasing lateral root formation in Arabidopsis [58]. UV-B light decreases organic acid exudation, although some species do not show this response [59]. A higher red/far-red ratio shifted the exudation profile in tomato to more primary metabolites [60]. Monochromatic red or blue light increased carbohydrate exudation in lettuce, and this effect was influenced by plant age and root environment. This response was observed at early growth stages and was quickly diminished afterward. Moreover, the response was only observed in lettuce grown in a substrate, but not in lettuce grown using the deep flow technique [35]. Remarkably, the regulation of lateral roots through far-red light also depended on root-environmental conditions, namely, nitrate [58]. These findings suggest that light quality modulates root exudation, although the specific effects depend on other factors, such as species and environmental conditions.

Elevated CO_2_ levels are commonly used in protected horticulture to boost plant growth. A recent meta-analysis summarized the effects of elevated CO_2_ levels on root exudation [44]. Elevated CO_2_ increased the exudation of total carbon, organic acids (such as citric acid and malic acid), and soluble sugars while decreasing amino acid exudation. This effect varies depending on a plant’s growth stage [61].

Temperature is often regulated in greenhouse settings to extend the growing season. Beech forests and coniferous forests showed increased root exudation with higher temperatures [62,63]. In a water culture, legumes exhibited an increase in total nitrogen exudation as temperatures rose [64]. A meta-analysis confirmed that elevated temperatures led to higher total carbon exudation rates, particularly for malic acid [44]. Temperature has a uniform effect on root exudation.

## 5. Conclusions

Root exudation in soilless systems generally results in a higher total C exudation rate compared to natural soils. Moreover, the exudation profile varies depending on the plant species, with different species showing distinct patterns of change. Root exudation decreases as plants age, and most environmental adaptations occur during the early growth stages. Several environmental factors associated with soilless cultivation like nutrient content play an important role: N increases amino acid exudation, P enhances organic acid exudation, and K reduces sugar exudation. Light also plays a key role; reduced light intensity and shading decreases the exudation of N-containing compounds. Increased artificial light intensity reduces total C exudation and the exudation of certain autotoxins in lettuce. Light quality affects root exudation rates and profiles as well, but these changes are highly species-specific and influenced by the root environment. Elevated CO_2_ levels increase the exudation of total C, organic acids, and soluble sugars while decreasing the exudation of amino acids. The impact of CO_2_ elevation depends on the plant growth stage. Finally, higher temperatures increase total C exudation rates and, particularly, the exudation of malic acid.

## 6. Future Directions

While some understanding exists regarding how soilless cultivation systems impact root exudation, many factors remain inadequately represented in the scientific literature. Further research is needed to explore how variations in nutrient solutions, specifically those with different EC or pH levels, influence root exudation (Figure 2). Additionally, the effects of varying nutrient solution compositions on root exudation warrant further investigation. The growing medium itself requires more attention to elucidate how its sorption capacity, particle size, and porosity affect root exudation. Research on deep flow technique (DFT) systems should be expanded to understand how changes in root architecture in these systems influence exudation processes. Aboveground conditions, including temperature and elevated CO_2_, have been explored due to concerns over climate change. However, the impact of (additional) artificial light remains under-researched, with a particular gap in the knowledge regarding the effects of the photoperiod on root exudation. For all these studies, root weight or root length should be used to normalize the data and prevent confounding factorsfrom interpreting the results. Integrating findings across these factors will be essential for understanding how PGPBs can be optimized for maximal efficacy in soilless systems.

## Figures and Tables

**Figure 1 plants-14-00479-f001:**
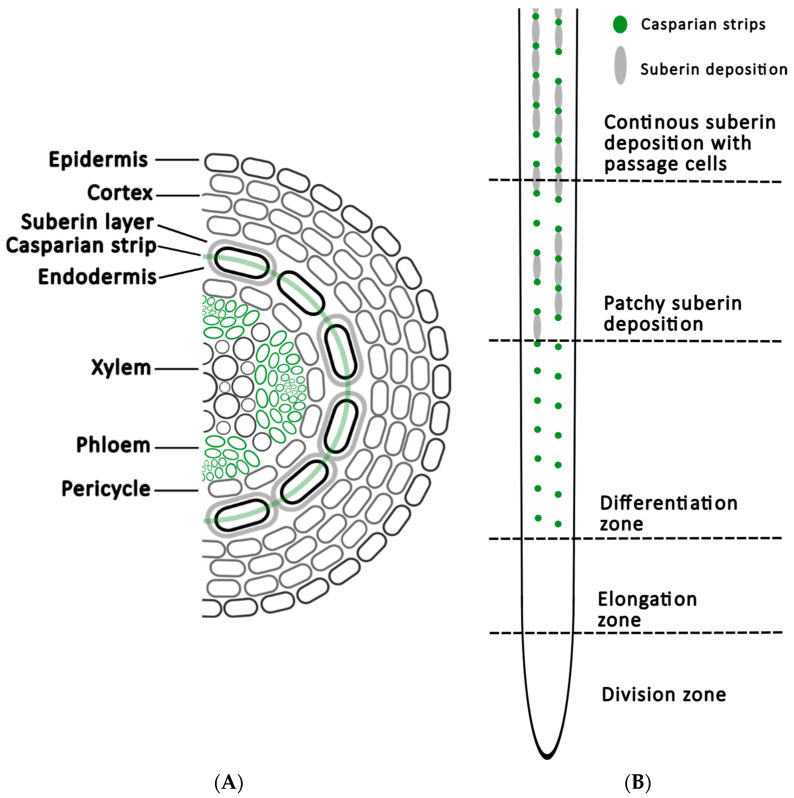
Schematic drawing based on Arabidopsis of (**A**) a cross-section of the root at the mature stage with continuous suberin deposition and (**B**) a transversal root section showing the different zones and locations of endodermal barriers.

**Figure 2 plants-14-00479-f002:**
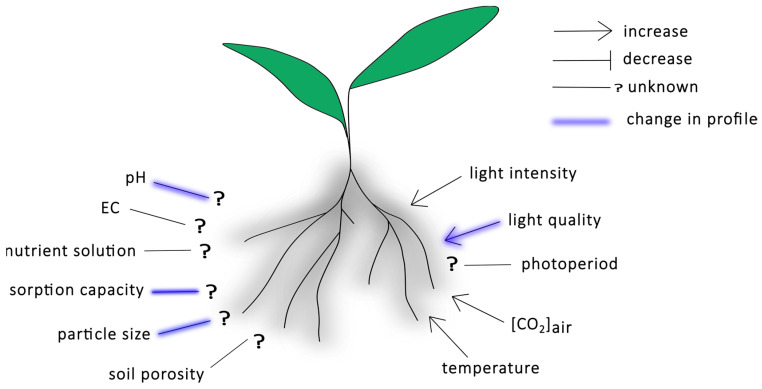
Key factors influencing root exudation in soilless culture systems.

## Data Availability

No new data were generated or analyzed in this study. Data sharing is not applicable to this article.

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
