# Peer review of "Root Exudates in Soilless Culture Conditions"

_plants, 2025, doi:10.3390/plants14030479_

Round 1

Reviewer 1 Report

Comments and Suggestions for Authors

The revision presents the major factors affecting the production of root exudates but there are no references to the root-microbiome variation as a result of that. Are there no studies on the variation of the microbiota under soilless conditions?

Reviewer 2 Report

Comments and Suggestions for Authors

This paper reviewed root exudates under soilless cultivation from various perspectives, including their composition, secretion mechanisms, and environmental factors affecting soilless systems, effectively linking this underground process with soilless cultivation, which holds scientific and application potential. Below are some suggestions for improvement:

  1. Emphasize the unique impacts of soilless cultivation. Compared to natural soil cultivation, soilless cultivation might have unique effects on root exudates, which should be the foundational starting point of this review. Unfortunately, no related descriptions were found in the current manuscript. The author should begin by elaborating on the distinctions between root exudates in soilless systems and natural soil environments.
  2. In line 16, the statement "Nutrient availability is a major driver" is unclear. The major driver of what? The author should specify its role clearly.
  3. In line 55, the text states: "In soilless horticulture, nutrient acquisition and allelopathy functions of exudate are useless, selection of (potentially beneficial) microbes is of interest to the grower." Why are the nutrient acquisition and allelopathy functions of exudates considered useless? The rationale behind this claim needs to be clarified to avoid confusion.
  4. In line 79, the key observation, "Total carbon (C) exudation rate is higher in hydroponic systems than in natural soils," lacks sufficient literature support. As an important conclusion, the author should provide more robust references and attempt to discuss the underlying mechanisms.
  5. Discuss mechanisms in detail. When mentioning data or experimental phenomena, the author could delve deeper into the mechanisms. For instance, exploring how different light qualities influence the compositional changes in root exudates through signal transduction pathways would enrich the discussion.

Reviewer 3 Report

Comments and Suggestions for Authors
